# Altered *N*-glycan composition impacts flagella-mediated adhesion in *Chlamydomonas reinhardtii*

**Nannan Xu**[1,2†]**, Anne Oltmanns**[3†]**, Longsheng Zhao**[4,5]**, Antoine Girot**[6]**, Marzieh Karimi**[6]**, Lara Hoepfner**[3]**, Simon Kelterborn**[7]**, Martin Scholz**[3]**, Julia Beißel**[3]**, Peter Hegemann**[7]**, Oliver Bäumchen**[6,8]**, Lu-Ning Liu**[4,9]**, Kaiyao Huang**[1]*****, Michael Hippler**[3,10]*****

[1]Key Laboratory of Algal Biology, Institute of Hydrobiology, Chinese Academy of Sciences, Wuhan, China; [2]University of Chinese Academy of Sciences, Beijing, China; [3]Institute for Plant Biology and Biotechnology, University of Münster, Münster, Germany; [4]Institute of Integrative Biology, University of Liverpool, Liverpool, United Kingdom; [5]State Key Laboratory of Microbial Technology, and Marine Biotechnology Research Center, Shandong University, Qingdao, China; [6]Max Planck Institute for Dynamics and Self-Organization (MPIDS), Göttingen, Germany; [7]Institute of Biology, Experimental Biophysics, Humboldt University of Berlin, Berlin, Germany; [8]Experimental Physics V, University of Bayreuth, Bayreuth, Germany; [9]College of Marine Life Sciences, and Frontiers Science Center for Deep Ocean Multispheres and Earth System, Ocean University of China, Qingdao, China; [10]Institute of Plant Science and Resources, Okayama University, Kurashiki, Japan

**\*For correspondence:**
huangky@ihb.ac.cn (KH);
mhippler@uni-muenster.de (MH)

†These authors contributed equally to this work

**Competing interests:** The authors declare that no competing interests exist.

**Abstract** For the unicellular alga *Chlamydomonas reinhardtii,* the presence of *N*-glycosylated proteins on the surface of two flagella is crucial for both cell-cell interaction during mating and flagellar surface adhesion. However, it is not known whether only the presence or also the composition of *N*-glycans attached to respective proteins is important for these processes. To this end, we tested several *C. reinhardtii* insertional mutants and a CRISPR/Cas9 knockout mutant of xylosyltransferase 1A, all possessing altered *N*-glycan compositions. Taking advantage of atomic force microscopy and micropipette force measurements, our data revealed that reduction in N-glycan complexity impedes the adhesion force required for binding the flagella to surfaces. This results in impaired polystyrene bead binding and transport but not gliding of cells on solid surfaces. Notably, assembly, intraflagellar transport, and protein import into flagella are not affected by altered *N*-glycosylation. Thus, we conclude that proper *N*-glycosylation of flagellar proteins is crucial for adhering *C. reinhardtii* cells onto surfaces, indicating that *N*-glycans mediate surface adhesion via direct surface contact.

## Introduction

*N*-glycosylation, as one of the major post-translational modifications, takes place along the ER/Golgi secretion route and consequently most *N*-linked glycans are found on proteins facing the extracellular space. Initial steps of *N*-glycosylation in the ER are highly conserved among most eukaryotes and consist of the synthesis of a common prebuilt *N*-glycan precursor onto a dolichol phosphate. Following the transfer of the glycan precursor onto the asparagine of the consensus sequence N-X-S/T of a nascent protein (where X can be any amino acid except proline), the glycoprotein is folded by the glycan recognizing chaperones Calnexin and Calreticulin (*Stanley and Taniguchi, 2017*). Subsequent

*N*-glycan maturation steps in the Golgi are species dependent and give rise to a high variety of *N*-glycan structures. In land plants, Golgi maturation leads to *N*-glycans modified with β1,2-core xylose and α1,3-core fucose (*Strasser, 2016*). In *Chlamydomonas reinhardtii*, a unicellular biflagellate green alga, *N*-glycans can be decorated with core xylose and -fucose (*Lucas et al., 2020*; *Oltmanns et al., 2019*; *Schulze et al., 2018*). Additionally, *6O*-methylation of mannose and addition of a terminally linked β1,4-xylose were reported (*Mathieu-Rivet et al., 2013*). While the functional advantage of *N*-linked glycans to mature proteins is hardly understood, it is known that blocking the synthesis of a full *N*-glycan precursor results in hypoglycosylated proteins that cannot be folded properly (*Gardner et al., 2013*). Instead, they are degraded via the ER-associated degradation pathway (ERAD) and are consequently not targeted correctly (*Adams et al., 2019*; *Cherepanova et al., 2016*). Therefore, impairment of glycosylation at such early stage is lethal in both uni- and multi-cellular organisms and only when inhibition of glycosylation is carefully dosed (e.g. by tunicamycin), immediate physiological effects caused by hypoglycosylation can be observed (*Kukuruzinska et al., 1987*). Looking at *C. reinhardtii*, treatment of vegetative cells with tunicamycin lead to an impaired flagellar adhesiveness, indicating that glycoproteins are crucial for adhesion and the subsequent onset of gliding (*Bloodgood et al., 1987*). However, whether this phenotype is linked to mistargeting of proteins due to hypoglycosylation and/or the lack of *N*-glycans on the flagella surface is unclear.

Whole-cell gliding is one of two flagella-based motilities in *C. reinhardtii* besides swimming (*Ishikawa and Marshall, 2011*; *Kozminski et al., 1993*; *Snell et al., 2004*). In principle, the cell adheres to a surface via its flagella, positioning them in a 180° angle and initiates gliding along the solid or semisolid surface into the direction in which one flagellum is pointing (designating it as leading flagellum) (*Bloodgood, 2009*). Interestingly, flagella not only bind to large solid surfaces, but they also bind to small, inert objects (e.g. polystyrene microbeads) that are moved along the flagellar membrane. While the two events, summarized as flagellar membrane motility, are believed to underly the same molecular machinery, it is assumed that they start with an adhesion of flagella membrane components to the surface (*Bloodgood and Salomonsky, 1998*). A micropipette force measurement approach recently showed that the flagella adhesion forces on different model surfaces with tailored properties lie in the range of 1 to 4 nN and that only positive surface charge diminished the adhesion force significantly (*Backholm and Bäumchen, 2019*; *Kreis et al., 2019*; *Kreis et al., 2018*). These findings imply that the adhesion system of *C. reinhardtii* has developed toward great flexibility instead of high specificity, in line with the high diversity of solid surfaces dwelled by the microalga in nature, ranging from soil and sand to wet leaves, moss, and bark (*Harris, 2009*). Remarkably, surface iodination experiments in the early 1980s revealed a single protein called flagellar membrane glycoprotein 1B (FMG-1B) as the main player mediating surface contact (*Bloodgood and Workman, 1984*). FMG-1B is exclusively located in the flagellar membrane and has a remarkable size of around 350 kDa (4389 amino acids) with a large extra-flagellar part (4340 amino acids) anchored in the membrane via a single predicted trans membrane helix of 22 amino acids (*Bloodgood et al., 2019*). As the name indicates, it is heavily *N*- and *O*-glycosylated. A recent knock down study showed, that it is the main constituent of the glycocalyx surrounding the flagellum. Additionally, a *fmg-1B* mutant showed a drastically reduced ability to glide (*Bloodgood et al., 2019*). Strikingly, FMG-1B is present at a high copy number and turns over rapidly within approximately 1 hr (*Bloodgood, 2009*). The rapid turnover is probably attributed to the fact that flagellar membrane components are constantly shed into the medium as flagellar ectosomes (*Bloodgood, 2009*; *Wood et al., 2013*). FMG-1B and another *N*-glycosylated membrane component, FAP113, have been shown to be eventually torn out of the membrane once bound to microbeads (*Kamiya et al., 2018*). Whether FAP113 is only involved in microbead binding or also in whole-cell gliding is unknown.

Recently, it was found that flagellar adhesion to surfaces is switchable by light, indicating that a blue-light photoreceptor signal is governing this process (*Kreis et al., 2018*). Following adhesion to the surface, a transmembrane signal mediates translation of the adhesion event into a calcium transient and protein phosphorylation cascade (*Bloodgood, 2009*; *Collingridge et al., 2013*; *Kreis et al., 2018*). According to the current model, an interaction of the short cytoplasmic part of FMG-1B with intraflagellar transport (IFT) may occur (*Laib et al., 2009*; *Shih et al., 2013*). IFT moves bidirectionally along the flagellar microtubules, the anterograde transport is driven by kinesin 2 and retrograde transport is driven by cytoplasmic dynein-1b (*Cole et al., 1998*; *Huangfu et al., 2003*;

*Kozminski et al., 1995*; *Lechtreck, 2015*; *Pedersen and Rosenbaum JLBT-CT in DB, 2008*; *Porter et al., 1999*; *Rosenbaum and Witman, 2002*). Since retrograde IFT trains pause relative to the adhesion site while FMG-1B tethers to the solid surface through its large extracellular carbohydrate domain (*Bloodgood, 2009*), the force generated by retrograde motor protein dynein-1b will push the microtubule into the opposite direction, dragging the cell body and the second flagellum behind; the gliding process is initiated (*Shih et al., 2013*).

Due to the high N-glycosylation level of the extraflagellar domain of FMG1-B, interacting with the solid surface, it was suggested that N-glycosylation could be crucial for adhesion, beyond proper glycoprotein folding. Therefore, we compared flagellar membrane motility of different mutant strains impaired in *N*-glycan maturation (characterized in *Schulze et al., 2018*). We found that *N*-glycan maturation indeed impacts the interaction of flagellum and surface in all mutants analyzed.

## Results

### Altered *N*-linked glycans do not change the flagellar localization of FMG-1B

To test whether *N*-glycan maturation in Golgi is important for flagellar surface motility in *C. reinhardtii*, two insertional mutants (IM) such as $IM_{Man1A}$, $IM_{XylT1A}$ and their double mutant $IM_{Man1A}xIM_{XylT1A}$ were studied. Initially, these mutants had been described in *Schulze et al., 2018*, where *N*-glycan patterns of supernatant proteins were analyzed and compared (*Figure 1A*). The insertional mutagenesis giving rise to these mutants was performed in the parental strain CC-4375 (a *ift46* mutant back-crossed with CC-124) complemented with IFT-46::YFP, referred to as WT-Ins throughout the current study. The first mutant, deficient in xylosyltransferase 1A ($IM_{XylT1A}$), produces *N*-glycans devoid of core xylose while simultaneously having a reduced length. Second mutant $IM_{Man1A}$, a knock down mutant of mannosidase 1A, is mainly characterized by a lack of 6*O*-methylation of mannose residues while the *N*-glycan length is slightly greater than in WT-Ins. Furthermore, *N*-glycans of $IM_{Man1A}$ are slightly reduced in terminal xylose and core fucose. Finally, a double mutant of the above two single mutants ($IM_{Man1A}xIM_{XylT1A}$, obtained by genetic crossing) produces *N*-glycans devoid of 6*O*- methylation but of WT-Ins length and carrying core xylose and fucose residues (*Figure 1A*). It is of note that in none of the mutants the flagellar length is altered as compared to WT-Ins (*Figure 1—figure supplement 1*). To confirm that flagellar *N*-glycan patterns of these mutants deviate from WT-Ins, whole-cell extracts and isolated flagella were probed with anti-HRP, binding to β1,2-xylose and α1,3-fucose attached to the *N*-glycan core (*Kaulfürst-Soboll et al., 2011*). In line with previous publications, the antibody showed a higher affinity toward *N*-glycoproteins synthesized by $IM_{Man1A}$ and $IM_{Man1A}xIM_{XylT1A}$ while it showed a decreased affinity toward probes of $IM_{XylT1A}$ (*Figure 1—figure supplement 2*; *Schulze et al., 2018*). Further lectin-affino blotting with concanavalin A (ConA) was performed on whole-cell extracts, revealing increased ConA-affinity in all three *N*-glycosylation mutants compared to WT-ins (*Figure 1—figure supplement 3*).

Since FMG-1B is the major constituent of the flagellar glycoproteome and to date the only protein proven to be involved in flagellar surface motility, different monoclonal antibodies raised against FMG-1B were employed to analyze FMG-1B localization (*Bloodgood et al., 1986*; *Long et al., 2016*). The whole-cell extracts or isolated flagella from $IM_{Man1A}$, $IM_{XylT1A}$, and their double mutant $IM_{Man1A}xIM_{XylT1A}$ were probed separately with the glycan epitope recognizing antibody or the antibody against the protein backbone of FMG-1B. FMG-1B was found in whole cells and flagella of WT-Ins and all three mutants. Hereby, the protein amount was similar in all four strains as indicated by the use of the FMG-1B protein-specific antibody (*Figure 1B*). Contrarily, the antibody raised against FMG-1B glycan epitopes barely detected whole-cell or flagella probes of $IM_{Man1A}$ and $IM_{Man1A}xIM_{XylT1A}$. Also, the FMG-1B glycan signal decreased in the mutant $IM_{XylT1A}$, particularly in the whole-cell sample. Taken together, these immuno-blots confirm that the *N*-glycan pattern of FMG-1B is altered in the mutants, while the protein localization is not affected by this alteration. In addition, label-free mass spectrometric quantification confirmed that FMG-1B is correctly targeted to the flagella in all mutants analyzed (*Figure 1C*). The same is true for FAP113, another protein shown to be involved in flagella surface motility (*Figure 1D*).

To compare the localization of glycan and protein of FMG1-B in WT-Ins and mutants, the cells were immuno-stained with the two FMG-1B-specific antibodies described above. A uniform signal of

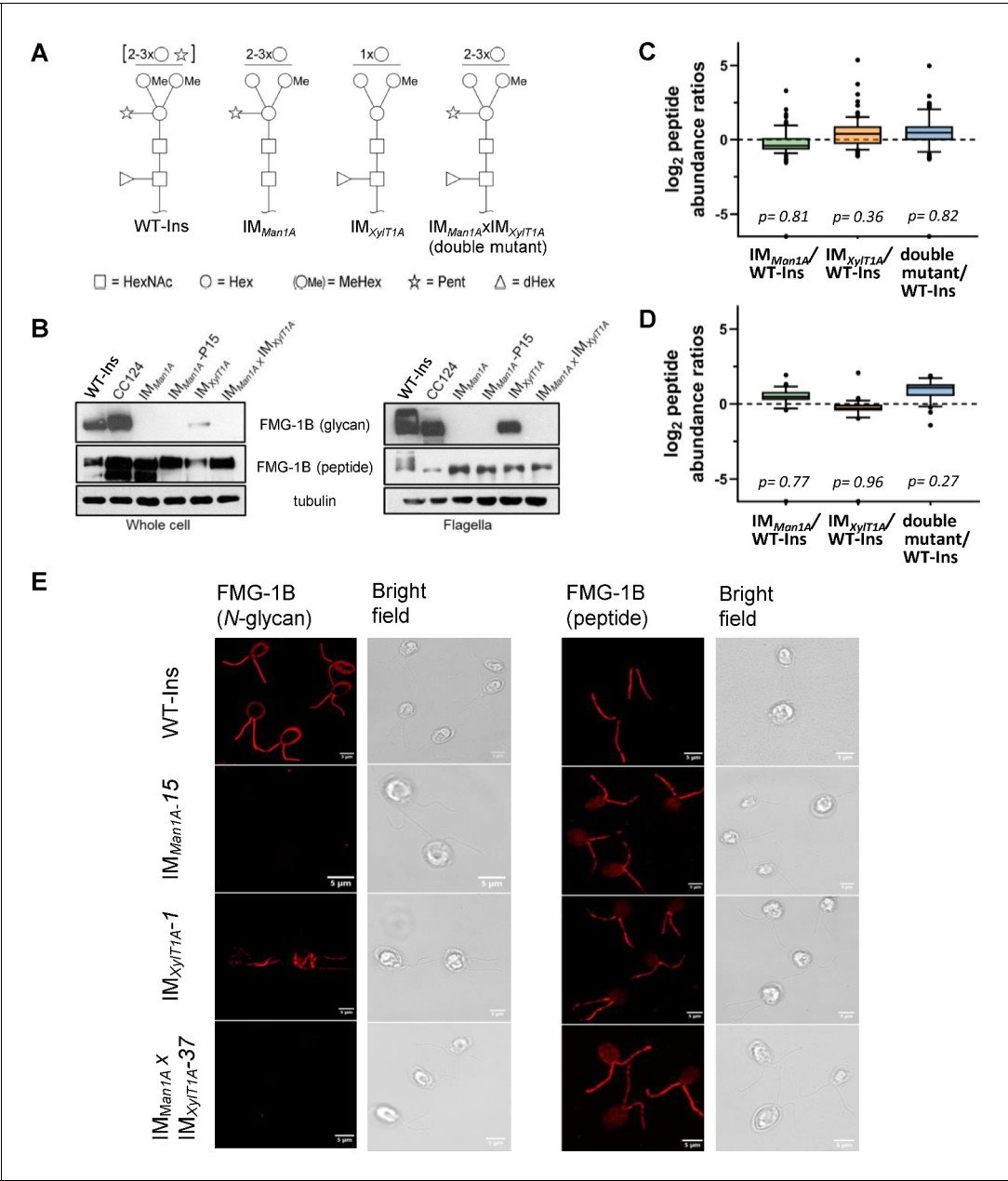

*Figure 1 continued*

**Figure supplement 5.** Genetic crossing of original IM strains (mt+) with CC124 (mt-) to obtain mutants lacking IFT46::YFP.

**Figure supplement 6.** The altered *N*-glycan did not change the localization FMG-1B in flagella.

**Figure supplement 7.** Quantitative mass spectrometry of isolated flagellar from WT-Ins and *N*-glycosylation mutants.

the glycan-specific antibody was found in the flagella and the cell wall of WT-Ins (*Figure 1E*, left panel). It should be noted, that such cross reaction with cell wall localized glycosylated proteins has been reported previously (*Bloodgood et al., 1986*). In line with the immuno-blotting experiment, no glycan signal was observed in the flagella or cell wall of IM$_{Man1A}$ and the double mutant, while signal intensity was low in the flagella of mutant IM$_{XylT1A}$ (*Figure 1E*, left panel and). The faint FMG-1B signal in IM$_{XylT1A}$ compared to the WT-Ins was clearly observed when WT-Ins and IM$_{XylT1A}$ were mixed prior to immuno-staining (*Figure 1—figure supplement 6*). When the FMG-1B peptide antibody was used, a uniform signal is found in the flagella of WT-ins, M$_{Man1A}$, IM$_{XylT1A}$, and the mutant IM$_{Man1A}$xIM$_{XylT1A}$ (*Figure 1E*, right panel). In summary, these data show that altered *N*-glycan maturation did not affect the flagellar localization of FMG-1B. This is in line with early findings by Bloodgood et al., reporting proper FMG-1B targeting to the flagella in a *N*-glycosylation mutant termed L23 showing increased ConA-affinity (*Bloodgood et al., 1987*). Although FMG-1B is the most prominent and best studied flagella membrane glycoprotein, there might be other glycoproteins involved in flagellar adhesion. Nevertheless, no protein was found consistently and significantly changed in abundance in flagella of the IM strains analyzed when compared to WT-Ins (*Figure 1—figure supplement 7*), also indicating that flagellar assembly is not considerably altered in the mutants versus WT.

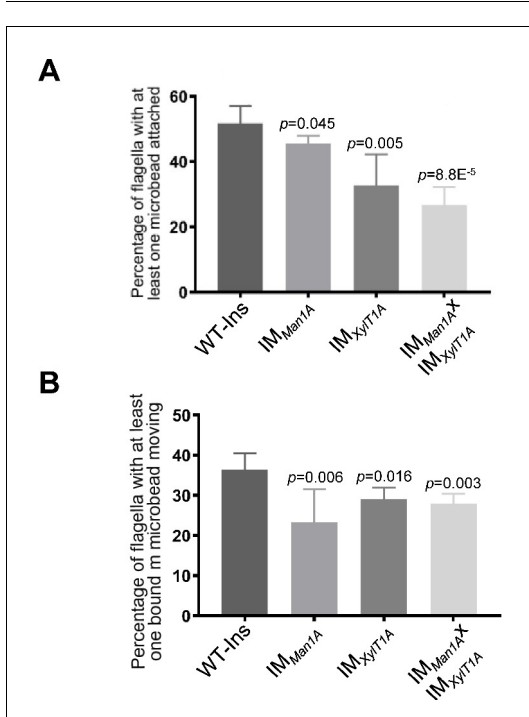

**Figure 2.** Altered *N*-glycosylation diminishes flagellar polystyrene bead attachment and -transport. (A) Percentage of flagella with at least one polystyrene bead bound. Cells were incubated with polystyrene beads (0.7 µm in diameter) and subsequently analyzed by light microscopy. (B) Percentage of polystyrene beads transported along the flagellum with at least one polystyrene bead bound. Results present mean of three replicates with 50 cells analyzed per replicate. Error bars show SEM of three replicates. T-test was used for statistical analysis. Source data can be found in *Figure 2—source data 1*.

The online version of this article includes the following video and source data for figure 2:

**Source data 1.** Raw data of attachment and movement of microbeads to and along flagella.

**Figure 2—video 1.** Attachment and movement of a microsphere to and along flagella.

https://elifesciences.org/articles/58805#fig2video1

## Altered *N*-linked glycans attenuate bead attachment to and movement along the flagellar membrane

In *C. reinhardtii*, polystyrene microspheres adhere to and move bidirectionally along the flagellar surface (*Bloodgood, 1981*). To check the effect of altered *N*-glycans on these processes, beads with a diameter of 0.7 µm were added to cell suspensions of IM$_{Man1A}$, IM$_{XylT1A}$, and the double mutant and the number of beads attached to- or moved along flagella were quantified (*Figure 2A and B*, exemplary video in *Figure 2—video 1*). In WT-Ins strains, about 52% of all flagella had at least one bead attached, whereas the percentage of flagella with beads bound decreased to 46% in IM$_{Man1A}$, 33% in IM$_{XylT1A}$ and 27% in the double mutant (*Figure 2A*). Among the beads attached, 37% moved along flagella of WT-Ins, 22% in IM$_{Man1A}$, 29% in IM$_{XylT1A}$ and 27% in the double mutant (*Figure 2B*). Source data for *Figure 2* can be

found in *Figure 2—source data 1*. These data suggested that interaction of flagellar membrane and surface is altered due to altered *N*-glycan composition in these mutants.

## Quantification of the flagella-mediated adhesion using atomic force microscopy

Flagella-mediated surface adhesion force was measured via atomic force microscopy (AFM) (*Figure 3A*). Here, cells adhered to a cover slide were attached to an AFM cantilever via physical contact (*Liu et al., 2011*). Subsequently, the AFM cantilever was pulled upwards and the force required to pull the cells was recorded (*Figure 3B*). To inhibit whole-cell gliding during the measurement, ciliobrevin D was used to inhibit dynein-1b activity and consequently the cell gliding (*Firestone et al., 2012*). Remarkably, the forces necessary to overcome the adhesion of *C. reinhardtii* flagella to the surface were significantly reduced in these three mutants analyzed as compared to WT-Ins (*Figure 3B and C*). Especially in the double mutant the adhesion force was reduced from 8 nN in WT-Ins to 1 nN, while the average energy was reduced from 4 to 0.5 J nm$^{-1}$ (*Figure 3C and D*). Source data can be seen in *Figure 3—source data 1*. This result indicates that an altered *N*-glycan composition impacts the flagellar adhesion force onto a solid substrate.

## Quantification of flagellar adhesion using a micropipette force measurement approach

To validate the AFM adhesion force measurements, an independent in vivo force measurement approach (*Backholm and Bäumchen, 2019*; *Kreis et al., 2018*) was used with another genetic background mutant, xylosyltransferase 1A (CRISPR$_{XylT1A\_1}$), generated in parental wildtype SAG11-32b (WT-SAG) by employing CRISPR/Cas9 (*Figure 4—figure supplement 1*). As a wavelength dependency of flagellar adhesion had been revealed using this approach (*Backholm and Bäumchen, 2019*), adhesion forces of the same cells were measured under precisely controlled blue- and red-light conditions using a micropipette adhesion force measurement approach. Adhesion forces were measured in presence and absence of dynein-1b inhibitor ciliobrevin D. Importantly, adhesion forces in CRISPR$_{XylT1A\_1}$ were significantly diminished in comparison to respective WT under both ciliobrevin D conditions confirming AFM results (*Figure 4*). Illuminating cells with red light dramatically decreased the adhesion force in both WT-SAG and CRISPR$_{XylT1A\_1}$ in presence as well as in absence of ciliobrevin D. Notably, removal of ciliobrevin D resulted in a significant decrease in adhesion force for both WT-SAG and CRISPR$_{XylT1A\_1}$ from ~2.6 nN to ~1.3 nN in WT-SAG and from ~1.8 nN to ~1 nN in CRISPR$_{XylT1A\_1}$. Source data can be seen in *Figure 4—source data 1*.

## The effect of altered *N*-glycosylation on IFT and gliding

As presented in *Figure 3*, the strongest effect on adhesion forces assessed was observed in the double mutant IM$_{Man1A}$xIM$_{XylT1A}$ when compared to WT-Ins. In the absence of ciliobrevin D, the adhesion force measured by AFM in mutant IM$_{Man1A}$xIM$_{XylT1A}$ is still significantly lower than in WT-Ins (*Figure 5A*). This is in line with the WT-SAG and CRISPR$_{XylT1A\_1}$ micropipette adhesion force measurement performed in the absence of ciliobrevin D, where also CRISPR$_{XylT1A\_1}$ had a lower adhesion force (*Figure 4*). Interestingly, addition of ciliobrevin D resulted in significantly increased adhesion forces as seen for WT-SAG or WT-Ins (*Figure 4* and *Figure 5A*). Taken together, these results suggested that active dynein-1b might reduce surface adhesion forces. On the other hand, it implied that IFT might be hampered via altered *N*-glycosylation as surface adhesion forces were smaller in the *N*-glycosylation mutants. Therefore, IFT velocity and gliding ability of IFT46::YFP expressing WT-Ins and IM$_{Man1A}$xIM$_{XylT1A}$ in absence of ciliobrevin D were assessed by using total internal reflection fluorescence (TIRF) microscopy. Videos of adhered cells generated by TIRF microscopy were evaluated manually with help of kymographs in Fiji software (*Figure 5B*). Obtained data revealed that neither the proportion of gliding events (gliding velocity higher 0.3 µm*s$^{-1}$), nor gliding speed distribution was significantly diminished when comparing WT-Ins and the double mutant (*Figure 5C*). Likewise, anterograde and retrograde IFT velocities were found at WT-Ins values when comparing WT-Ins and the double mutant of adherent cells (*Figure 5C*), implying no significant impact of altered *N*-glycan maturation on IFT. Source data for *Figure 5* can be seen in *Figure 5—source data 1*. Lastly, to rule out the possibility that ciliobrevin D might result in elevated adhesion forces due to a toxic side effect, we generated the triple mutant dynein-1b$^{ts}$x IM$_{Man1A}$ x IM$_{XylT1A-4-}$

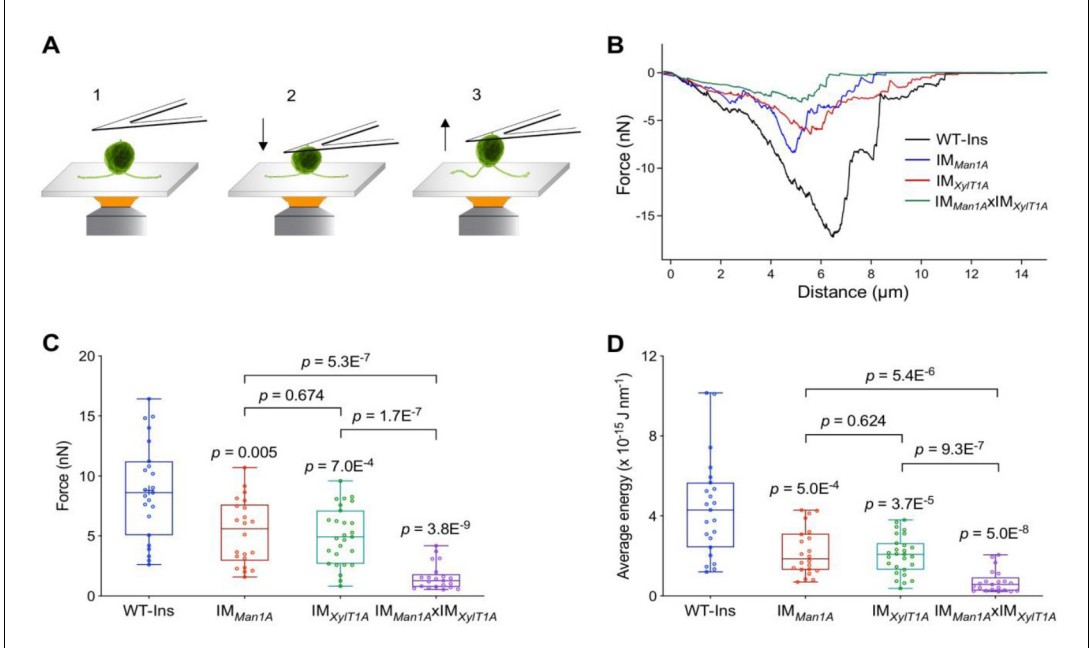

**Figure 3.** Quantification of the flagella-mediated adhesion using atomic force microscopy. (A) Diagram of experimental procedures for force measurement: the cell adhered to the surface (1); the cell attached to the AFM cantilever (2); the cell was pulled up from the surface by AFM cantilever (3). Please note, that retrograde IFT, i.e. gliding was inhibited by ciliobrevin D during all measurements presented here. (B) Representative force curves acquired for strains including WT, IM$_{Man1A}$, IM$_{XylT1A}$, and IM$_{Man1A}$xIM$_{XylT1A}$. (C) Flagella adhesion forces of WT-Ins, IM$_{Man1A}$, IM$_{XylT1A}$, and IM$_{Man1A}$xIM$_{XylT1A}$ were generated from force curves (B). (D) Average energy of flagellar adhesion of WT-Ins, IM$_{Man1A}$, IM$_{XylT1A}$, and IM$_{Man1A}$xIM$_{XylT1A}$. Three biological replicates were performed with minimum 5 cells measured per replicate. The p-values are obtained from a two-sided, two sample t-test of mean values. Source data can be seen in *Figure 3—source data 1*.

The online version of this article includes the following source data and figure supplement(s) for figure 3:

**Source data 1.** Raw data of AFM mesurement (force, average energy).
**Figure supplement 1.** Analysis of the force and energy required to overcome the adhesion of *C. reinhardtii* flagella to the surface from AFM force curves.
**Figure supplement 1—source data 1.** Exemplary data of AFM measurement.
**Figure supplement 2.** Detachment distance and total energy of the flagella adhesion quantified by atomic force microscopy.
**Figure supplement 2—source data 1.** Raw data of AFM measurement (detachment distance, total energy).

13# by crossing IM$_{Man1A}$xIM$_{XylT1A}$ with CC-4423 (*Figure 5—figure supplement 1A–C*). CC-4423 is characterized by the expression of a temperature sensitive transcript of dynein-1b leading to a depletion of dynein-1b at restrictive temperatures followed by an attenuation of retrograde IFT and flagella disassembly (*Engel et al., 2012*). Subsequently, adhesion forces were measured via AFM at restrictive temperatures, that is, under conditions mimicking the ciliobrevin D dependent inactivity of dynein-1b. As it cannot be excluded that flagella shortening, as induced upon temperature shift (*Figure 5—figure supplement 1D*), might impact flagellar adhesion forces, the triple mutant treated with 20 mM NaPPi was assessed by AFM as control. Indeed, it was found that adhesion forces differed when measured at a defined flagella length of about 6.5 µm: while control cells showed an average adhesion force of 0.48 nN, cells depleted from dynein-1b showed a significantly elevated adhesion force of 1.64 N. Importantly, NaPPi only induces flagellar shortening (*Figure 5—figure supplement 1E*) but does not affect dynein-1b (*Dentler, 2005*), therefore, differences observed are directly correlated to the action of dynein-1b. Thus we suggest, that the action of dynein-1b reduces flagellar adhesion forces due to a destabilization of surface adhered protein clusters.

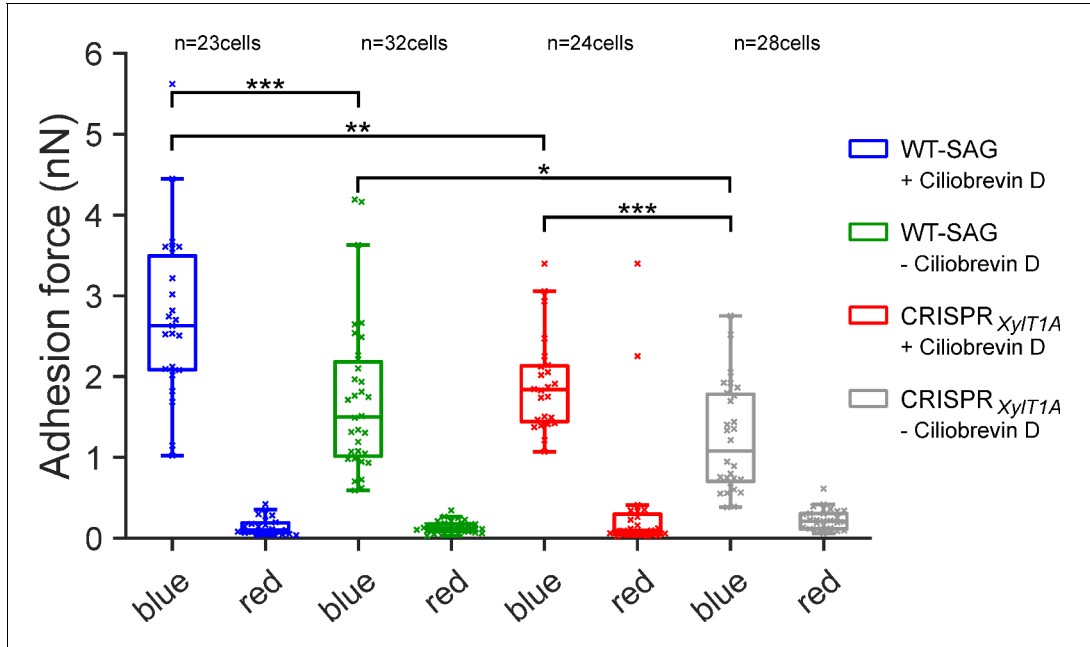

**Figure 4.** Assessing flagella adhesion forces using micropipette force microscopy. Flagella-mediated adhesion forces acquired for WT-SAG and a xylosyltransferase 1A mutant generated in the genetic background of WT-SAG (CRISPR$XylT1A\_1$). Micropipette force measurements of the same cells were performed for both strains under blue and red light in the (+) presence or (-) absence of ciliobrevin D. Mean values of 10 measurements per cell are depicted, statistical analysis was performed on mean values. The p-values obtained a from Kolmogorov-Smirnov test are respectively (from top to bottom): (***) p=0.0004, (**) p=0.0016, (*) p=0.0491, (***) p=0.0009. Source data can be seen in *Figure 4—source data 1*.
The online version of this article includes the following source data and figure supplement(s) for figure 4:

**Source data 1.** Raw data of micropipette force measurement.
**Figure supplement 1.** Xylosyltransferase 1A mutant generated via CRISPR/Cas9 supports findings in IM$_{XylT1A}$.
**Figure supplement 2.** Immunoblot proving the presence of FMG-1B in flagella of CRISPR$_{XylT1A.1}$ and CRISPR$_{XylT1A.2}$.
**Figure supplement 3.** Study of the effect of DMSO on the flagella adhesion forces using micropipette force microscopy.

## Discussion

Our data revealed that the maturation of *N*-glycans has an impact on flagella-mediated cell adhesion in *C. reinhardtii.* At the same time, IFT and gliding velocity were not changed due to altered *N*-glycosylation.

Microbead binding was found diminished in IM strains, implying that the flagellar surface has an altered affinity toward microbeads. In line with this, their surface adhesion forces were significantly reduced compared to WT-Ins. The AFM data were confirmed by assessing another XylT1A mutant created via CRISPR/Cas9, using micropipette force measurements. It should be noted that *N*-glycan patterns of IM$_{XylT1A}$ and CRISPR$_{XylT1A}$ were comparable and thereby strengthen the proposed role of XylT1A as core xylosyltransferase (*Lucas et al., 2020*; *Schulze et al., 2018*). Forces measured for WT strains and *N*-glycosylation mutants with AFM and micropipette force measurement confirmed that differential *N*-glycan maturation, i.e. altered *N*-glycan structures attached to mature proteins, lowers the adhesion force of flagella to a surface. These changes in adhesion forces were not accompanied by consistent drastic changes in the flagellar proteomes. For example, FMG-1B and FAP113, to date the only two known proteins involved in surface adhesion, were found in comparable amounts in WT-Ins and mutants (*Figure 1* and *Figure 4—figure supplement 2*; *Bloodgood et al., 2019*; *Kamiya et al., 2018*). Of note, also FMG-1A is localized in flagella and its abundance was unaltered between WT and mutants in vegetative cells (*Figure 1—figure supplement 4*). This contrasts the current assumption that FMG-1A is solely expressed in reproductive cells and opens the question whether it might have a similar role as FMG-1B, given the high similarity of the two proteins (*Bloodgood, 2009*). Interestingly, gliding of mutant strains on solid surface was not affected. The current model for flagella-mediated cell adhesion and subsequent gliding proposes that the

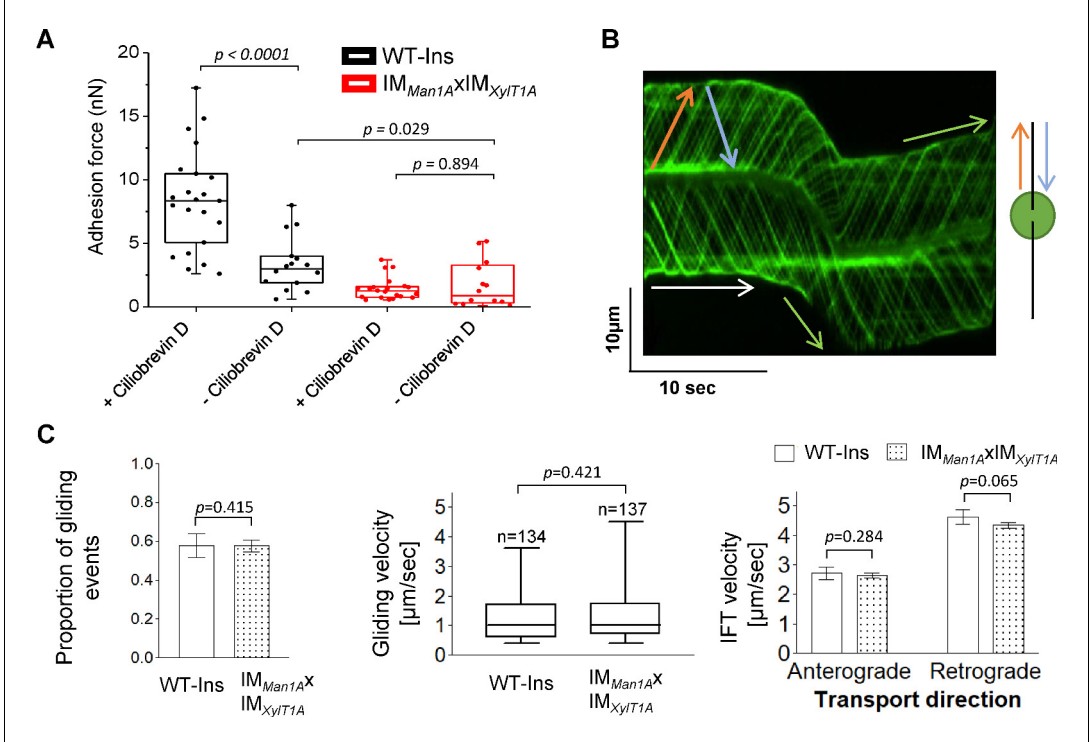

**Figure 5.** IFT and gliding are unaffected in IM$_{Man1A}$xIM$_{XylT1A}$. **(A)** Adhesion forces acquired for WT-Ins and the double mutant IM$_{Man1A}$xIM$_{XylT1A}$ in the absence or presence of ciliobrevin D via AFM, respectively. Three biological replicates were performed with minimum 5 cells measured per replicate. **(B)** Representative kymograph of the movement of IFT46::YFP in flagella of WT-Ins acquired with TIRF microscopy used to calculate the velocity of gliding and IFT. White arrow: non-gliding event (v < 0.3 μm*s$^{-1}$); green arrow: gliding event (v > 0.3 μm*s$^{-1}$); red arrow: anterograde IFT track; blue: retrograde IFT track. **(C)** IFT and gliding are not significantly altered in the double mutant compared to WT-Ins. Proportion of gliding events (left), gliding velocity (excluding non-gliding events; middle), IFT velocities in either direction (right). Three biological replicates were performed with 10 cells evaluated per replicate corresponding to 300 IFTs/replicates/strain in case of IFT velocity. Error bars in bar plots represent SD of three replicates. Student t-test was performed comparing mean values of replicates in regard of proportion of gliding events and IFT velocity. Distribution of gliding velocities was analyzed by use of Mann-Whitney U test, n represents number of gliding events measured. Gliding and IFT analysis has been performed in absence of ciliobrevin D. Source data for can be seen in *Figure 5—source data 1*.

The online version of this article includes the following source data and figure supplement(s) for figure 5:

**Source data 1.** Raw data of IFT tracking and gliding velocity by TIRF microscopy.

**Figure supplement 1.** Adhesion force increases in temperature sensitive dhc1-b mutant at restrictive temperature.

**Figure supplement 1—source data 1.** Raw data of AFM measurement proving the influence of dynein -1b.

extracellular part of certain glycoproteins such as FMG-1B adheres to the surface, cytoplasmic moieties of these proteins are bound to ongoing retrograde IFT directly or indirectly upon calcium- and light dependent stimulus which is followed by an onset of gliding (*Kreis et al., 2018*; *Shih et al., 2013*). Assuming that altered *N*-glycan maturation does not impact initial protein folding in the ER (as those steps are spatially and temporally separated), our data revealed that changed *N*-glycosylation, did not alter targeting of respective glycoproteins to flagella nor the velocity of IFT. Thus, changes in surface adhesion are likely linked to *N*-glycoprotein epitopes and their direct interaction with the solid or semisolid surface. How specific *N*-glycan moieties modulate adhesion force is subject of future research, particularly considering the finding that flagella-mediated adhesion of *C. reinhardtii* has been shown to be largely unaffected by different substrate surface properties (*Kreis et al., 2019*).

Notably, IFT and gliding were not changed between double mutant and WT-Ins (*Figure 5*). The fact that altered *N*-glycosylation diminished the force of cells to adhere to surfaces but did not affect IFT, strongly suggests that adhesion to surfaces and IFT are not necessarily coupled. As discussed below, adhesion probably evolved independently of the necessity to enable cell gliding.

Moreover, it can be concluded that *N*-glycosylation does not significantly impact differences in light perception, as the adhesion forces for WT-SAG and CRISPR$_{XylT1A}$ were significantly stronger under blue than under red light (*Figure 4*).

In summary, taking advantage of single-cell adhesion force measurements, our data revealed that cell adhesion was significantly impaired in *C. reinhardtii N*-glycosylation mutant strains. Our data further suggested that flagellar assembly, IFT and FMG-1B transport into flagella were not affected by altered N-glycosylation implicating no role of N-glycosylation in these processes. Instead, proper N-glycosylation of flagellar proteins is crucial for adhering C. reinhardtii cells onto surfaces. Our observations further suggest that the remaining adhesion force, although diminished in N-glycan mutants, is still sufficient for gliding. Given the response of flagellar adhesion to blue light, it could potentially link adhesion to photo-protection which is also blue-light mediated, as adhesion might result in photoprotection via biofilm formation, which in turn would enable mutual cell shading (*Kreis et al., 2018*; *Petroutsos et al., 2016*).

## Materials and methods

### Culture growth
Cells were grown photoheterotrophically in tris-acetate-phosphate (TAP) medium under constant illumination at 50 μmol photons*s$^{-1}$*cm$^{-2}$ unless stated otherwise.

### Measurement of flagellar length and flagellated cells
The cells were fixed with 0.5% Lugol's solution for 1 hr at room temperature. Flagellar length measurements were performed using a phase microscope (Nikon Eclipse Ti) equipped with an electron multiplying charged-coupled device. For each sample, at least 50 flagella were measured. For the measurement of flagellated cells, at least 100 cells were counted for each strain in biological triplicates.

### Generation and analysis of a CRISPR/Cas9 mutant strain
Mutagenesis was performed on the WT strain SAG11-32b following the protocol described in *Greiner et al., 2017* employing the transformation of a pre-built Cas9:guideRNA complex (Cas9 target seuence in the XylT1A gene: ACGAACACCCCAACACCAAT) simultaneously with a plasmid encoding for a paromomycin resistance via electroporation. Following selection with paromomycin, putative mutants were screened by PCR using the primer pairs short_fw: TACAAAGAACGGGACG-CAGG, short_rev: CATTGAAGCTCATCCAGACAC and long_fw: AAGGGTCACGGCACGGTATG, long_rev: CCTGAAGCACCCATGATGCACG. Genomic *XylT1A* regions of candidate strains showing not-WT like band patterns were sequenced. In total, two mutant strains differing in the DNA inserted following the Cas9 cutting site were identified (CRISPR$_{XylT1A\_1}$ and CRISPR$_{XylT1A\_2}$). Next, XylT1A protein levels were quantified by parallel reaction monitoring (PRM) and supernatant *N*-glycan compositions were assessed by IS-CID mass spectrometry. Additionally, flagella were isolated, separated by SDS-PAGE and, after transferred to a nitrocellulose membrane, probed with the protein backbone FMG-1B-specific antibody.

### Flagella isolation
Flagella isolation from cultures in the mid-log growth phase was performed as described elsewhere by the pH shock method (*Witman et al., 1972*). Pellets containing flagella samples were stored at −80°C until further use for immunoblotting or sample preparation for mass spectrometric measurements.

### Immunoblotting
Frozen, dry flagella and whole-cell samples were resuspended in lysis buffer (10 mM Tris/HCl, pH = 7.4, 2% SDS, 1 mM Benzamidine and 1 mM PMSF) and subjected to sonication for 10 min. After pelleting cell debris, the protein concentration was determined using the bicinchoninic acid assay (BCA Protein Assay Kit by Thermo Scientific Pierce). Volumes corresponding to 30 μg of protein were separated by SDS-PAGE, transferred to nitrocellulose membranes and incubated with antibodies as indicated.

## Lectin-affino blotting with concanavalin A and HRP

Frozen, dry whole-cell samples were resuspended in lysis buffer (10 mM Tris/HCl, pH = 7.4, 2% SDS, 1 mM Benzamidine and 1 mM PMSF) and subjected to sonication for 10 min. After pelleting the not-soluble cell debris, the protein concentration was determined using the bicinchoninic acid assay (BCA Protein Assay Kit by Thermo Scientific Pierce). Volumes corresponding to 50 µg of protein were separated by SDS-PAGE and transferred to nitrocellulose membrane. Membrane was incubated with ConA (1 µg/mL in TBST + 1 mM CaCl$_2$ + 1 mM MnCl$_2$) for 1.5 hr at room temperature. Subsequently membrane was washed and incubated 1 hr with HRP (5 µg/mL in TBST + 1 mM CaCl$_2$ + 1 mM MnCl$_2$). Excess HRP as well as Ca$^{2+}$ and Mn$^{2+}$ were removed by three washing steps with TBST, before affino blot was development via ECL.

## Sample preparation for mass spectrometric measurements

Frozen, dry flagella and whole-cell samples were treated as described in Immunoblotting. Volumes corresponding to 60 µg of protein were tryptically digested and desalted as described elsewhere (*Rappsilber et al., 2007*).

## Mass spectrometry measurements

Tryptic peptides were reconstituted in 2% (v/v) acetonitrile/0.1% (v/v) formic acid in ultrapure water and separated with an Ultimate 3000 RSLCnano System (Thermo Scientific). Subsequently, the sample was loaded on a trap column (C18 PepMap 100, 300 µm x 5 mm, 5 mm particle size, 100 Å pore size; Thermo Scientific) and desalted for 5 min using 0.05% (v/v) TFA/2% (v/v) acetonitrile in ultrapure water with a flow rate of 10 µL*min$^{-1}$. Following, peptides were separated on a separation column (Acclaim PepMap100 C18, 75 mm i.D., 2 mm particle size, 100 Å pore size; Thermo Scientific) with a length of 50 cm. General mass spectrometric (MS) parameters are listed in *Table 1*.

For quantification of glycosyltransferases, PRM (including a target list) was employed on whole-cell samples and respective spectra were analyzed with the Skyline software (*Pino et al., 2020*). For quantification of flagellar proteins, flagella samples were measured in biological quadruplicates in standard, not targeted, data dependent measurements. Following, peptide wise protein abundance ratios (IM/WT) were calculated with ProteomeDiscoverer (normalizing on a set of not membrane standing flagellar proteins) and filtered for proteins identified in at least 11 samples, for proteins having Abundance Ratio Adj. p-value<0.05 for at least one ratio and for proteins appearing in the flagellar proteome ChlamyFPv5 (*Pazour et al., 2005*).

In order to assign glycopeptides, samples were measured employing In-Source collision induced dissociation (IS-CID) as described previously followed by analysis of data with Ursgal and SugarPy (*Kremer et al., 2016*; *Oltmanns et al., 2019*; *Schulze et al., 2020*).

## Microbead measurements

Microbead binding- and transport assays were performed analogous to previous descriptions. (*Bloodgood et al., 2019*) Monodisperse polystyrene microspheres (0.7 µm diameter) were purchased from Polysciences, Inc Beads were washed with deionized water for three times and resuspended in NFHSM to make a store solution, which was used at 1:10 dilution in adhesion and motility detecting experiment.

To quantify the ability of bead binding, beads were added to 500 µL of cells at a density of $2 \times 10^7$ cells*mL$^{-1}$. After 5 min, cells were observed with a light microscope (Olympus, U-HGLGPS, 100X oil objective). A flagellum was scored as '+ bead' if beads adhered to it. The percentage of flagellar binding beads was calculated as: Percentage of flagellar binding beads = the number of '+ bead'/ (total number flagella scored) x 100%.

To obtain a kinetic measure of surface motility, cells were mixed with beads as above for 5 min and randomly observed under the light microscope. Each bead adhered to a flagellum was monitored for about 30 s. If beads moved along the flagella, we marked it as 'Moved bead' or it was 'Adhered bead'. The surface motility was calculated as: Percentage of moved beads along with flagella = 'Move bead' x/ ('Moved bead' + 'Adhered bead') 100%.

**Table 1.** MS parameters.

Relevant parameters used to acquire IS-CID and not fragmented TopN MS spectra as well as PRM data.

| | TopN without IS-CID | In-Source CID HCD | Parallel reaction monitoring | |
|---|---|---|---|---|
| Eluent compositions | Peptide trapping: 0.05% trifluoroacetic acid (TFA) in ultrapure water (A1), 0.05% TFA in 80% acetonitrile (B1) Peptide separation: 0.1% formic acid (FA) in ultrapure water (A2), 0.1% FA in 80% acetonitrile (B2) | | | LC parameters |
| Trap Column | C18 PepMap 100, 300 μM x 5 mm, 5 μm particle size, 100 Å pore size; Thermo Scientific | | | |
| Peptide trapping (eluents A1+B1) | 2.5% B1 at 5 μL/min for 5 min | | 2.5% B1 at 10 μL/min for 3 min | |
| Flow rate | 300 nL/min | | 250 nL/min | |
| Separation Column | Acclaim PepMap C18, 75 μm x 50 cm, 2 μm particle size, 100 Å pore size; Thermo Scientific | | | |
| Gradient for peptide separation (eluents A2+B2) | 2.5% B2 over 5 min, 2.5–45% B2 over 40 min, 45–99 % B2 over 5 min 99% B2 for 20 min 99–2.5% over 5 min 2.5% for 30 min | | 2.5% B2 over 5 min, 2.5–35% B2 over 105 min, 35–99 % B2 over 5 min 99% B2 for 20 min 99–2.5% over 5 min 2.5% for 40 min | |
| In-source CID | off | 80 eV | off | MS1 settings |
| Use lock masses | off | | on (m/z 445.12003) | |
| Resolution at *m/z* 200 (FWHM) | 70,000 | | | |
| Chromatographic peak width | 15 s | | | |
| AGC target | 3e6 | | | |
| Maximum injection time | 100 ms | | 50 ms | |
| Scan range | 600–3000 m/z | | 350–1600 m/z | |
| Mass tags | off | on | off | |
| TopN | 12 | | n/a | MS2 settings |
| Resolution at *m/z* 200 (FWHM) | 17,500 | | 35,000 | |
| Isolation window | 2 m/z | | 2 m/z (offset 0.5 m/z) | |
| AGC target | 1e5 | | | |
| Maximum injection time | 120 ms | | | |
| Normalized collision energy (NCE) | 30 | | 27 | |
| Minimum AGC target | 1.25e3 | | n/a | |
| Intensity threshold | 1e4 | | n/a | |
| Charge exclusion | unassigned,>5 | | n/a | |
| Dynamic exclusion | 15 s | | n/a | |

## AFM measurements

*C. reinhardtii* strains, grown in M1 medium under constant white illumination were grown for 65 hr, were allowed to adhere to a glass slide (immersed in ethanol for overnight, subsequently rinsed with MQ water) in fresh M1 medium for 15 min. Following, cells were incubated in the presence of ciliobrevin D for 1 hr (500 μL M1 supplemented with 200 μM ciliobrevin D). For AFM measurements, only adhered cells in gliding conformation having approximately similar appearance were analyzed. The MLCT-O10 AFM probe (Spring Const.: 0.03 N m$^{-1}$, length: 215 μm, width: 20 μm, resonant freq.: 15 kHz, Bruker) was soaked in acetone for 5 min, then subjected to UV illumination (distance to lamp: 3–5 mm) for 15 min. Then, the probe was immersed in 0.01% poly-l-lysine for 1 hr and afterwards rinsed with MQ water. Following, the probe was immersed with 2% glutaraldehyde for 1 hr and rinsed with MQ water before use. The AFM measurement was performed in Force Spectroscopy Mode in liquid at room temperature using a NanoWizard 3 AFM (JPK) equipped with a CellHesion stage (Z range: 100 μm) NanoWizard three head. The spring constant of the cantilever was routinely

calibrated using the contact-based thermal noise method. The AFM tip, modified as described, was lowered onto the cell surface at a rate of 10 µm s$^{-1}$ with a z scale of 25 µm. After contact, the applied force was maintained at 3 nN for 15 s. Then, the cell-attached probe was upraised at a rate of 1 µm s$^{-1}$. Force curves were processed with JPK SPM Data Processing (JPK). The forces and energy were determined as described in *Figure 3—figure supplement 1* (*Liu et al., 2011*). Three biological replicates were performed with minimum 5 cells measured per replicate.

## Micropipette force measurements

Cell culture growth and micropipette force measurements were performed following established recipes (*Kreis et al., 2019*; *Kreis et al., 2018*). In brief, *C. reinhardtii* strains WT-SAG and CRISP-R$_{XylT1A\_1}$ grew axenically in tris-acetate-phosphate (TAP) medium (Thermo Fisher Scientific) in a Memmert IPP 100Plus incubator on a 12 h day / 12 hr night cycle. The experimental approach is based on the use of a homemade micropipette force sensor, which allows for grasping a living cell by suction (*Backholm and Bäumchen, 2019*). The micropipette is calibrated by measuring the deflection induced by the weight of an evaporating water droplet at tip of the pipette. The adhesion force is obtained by bringing the flagella into contact with a piece of a silicon wafer (unilateral polished, Si-Mat) cleaned by sonication in ethanol, and by measuring the micropipette deflection during iterative approach and retraction of the substrate moving at 1 µm*s$^{-1}$. The substrate approach consists of pushing the cell such that the micropipette is deflected by 10 µm from the cell/substrate contact position, which is then followed by a dwell period of 10 s. The substrate is then retracted by 30 µm from the cell/substrate contact position at the same speed. The overall contact time between the flagella and the substrate is about 30 s. The illumination wavelength for blue and red light was 470 nm and 671 nm respectively, and realized by using narrow band pass interference filters (FWHM: 10 nm) added on top of the condenser of an inverted microscope (Olympus IX-73 and IX-83). During the adhesion force measurements, the cells were illuminated with a constant photon flux of 1019 photons*m$^{-2}$* s$^{-1}$ for both light conditions. For each cell, 10 adhesion force measurements were performed for each wavelength, whereby the order of red and blue light was varied randomly after five consecutive measurements.

In order to evaluate the influence of ciliobrevin D on the adhesiveness, a 200 µM stock solution of ciliobrevin D (Merck) was prepared in a 9:1 water: dimethylsulfoxide (DMSO, Purity: 99.9%, Sigma-Aldrich,) mixture. Then, 1.08 mL of this stock solution was added to 30 mL of culture to achieve a final concentration of 7 µM of ciliobrevin D in the cell suspension. The *C. reinhardtii* suspension containing ciliobrevin D was next incubated for 30 min and then centrifuged at 100 g for ten minutes, followed-up by a minimum of 30 min rest in the incubator. Finally, about 15 mL of the cell suspension was used to fill the liquid chamber. In parallel, a second suspension of *C. reinhardtii* cells was incubated using the same fraction of DMSO (but without ciliobrevin D), followed by the exact same experimental procedure to serve as a control group.

## TIRF imaging

Total internal reflection microscopy (TIRF) was applied to assess IFT and gliding behavior of *C. reinhardtii* strains expressing YFP-coupled IFT46. Therefore, cell densities were adjusted to 1 $\times$ 10$^5$ cells*mL$^{-1}$. Samples were loaded to a glass bottom microscopy chamber (µ-Slide 8 Well Glass Bottom) and refreshed every 20 min while imaging. TIRF microscopy was performed at room temperature with a Nikon Eclipse Ti and a 100x objective. IFT46::YFP was excited at 488 nm and fluorescence was recorded with an iXon Ultra EMCCD camera (Andor). For analysis, images were captured with NIS-Elements software over 30 s at 10 fps and a pixel size of 0.158 µm*pixel$^{-1}$. Images were evaluated by use of Fiji via manual evaluation of kymographs. Nett IFT velocities during gliding were calculated by subtracting corresponding gliding velocities. Three biological replicates were performed with 10 cells in gliding configuration analysed per replicate.

## Confocal imaging

Cells were incubated with primary antibodies (FMG-1B #8 and #61, available at dshb.com), subsequently incubated with a fluorescently labeled secondary antibody and analyzed by confocal microscopy as described previously (*Lv et al., 2017*). In brief, cells were plated on 1% poly (ethyleneimine) coated cover glass, decolorized and fixed in methanol at −20°C for 20 min, permeated cells in PBS

buffer for 1 hr, and then blocked in 5% BSA (Biosharp), 10% normal goat serum (Dingguo) and 1% fish gelatin (Sigma) in PBS. Incubated the samples with primary antibodies overnight, washed them, and incubated secondary antibody, washed the samples and mounted them on slides with nail polish. The slides were examined with a Leica confocal microscope (SP8). Images were acquired and processed by LAS X software (Leica) and ImageJ software. The 488 nm laser was used YFP excitation wavelength is 510 nm, the emission wavelength is 525 nm and the exposure time is 200 ms.

### Mating and Tetrad analysis

The plus and minus strains was incubated in 2 mL TAP-N medium ($2 \times 10^7$ cells/mL) under continuous light overnight for gametogenesis. 0.5 mL plus and minus gametes were mixed together and incubated for 2 hr for mating. Then 0.15 mL mixture was dispersed onto mature plate (4% agar). Plate was kept in dark for 5 days, then exposed to light for 24 hr. The unmated gametes were removed from the mature plate with razor blade and were killed using chloroform for 30 s. The agar contained about 30 zygotes was cut and transferred to a germination plate (1% agar). The plates were incubated in bright light till the spores were released from the zygote, then 100 µL water was added to the cut agar and was dispersed on whole plate. Single clones appeared within 3–5 days and were picked for further analysis.

### Materials and correspondence

The mass spectrometry proteomics data have been deposited to the ProteomeXchange Consortium (http://proteomecentral.proteomexchange.org) via the PRIDE partner repository with the dataset identifier PXD018353 and will be publically available upon acceptance of the manuscript (*Perez-Riverol et al., 2019*). For further requests, please contact K. Huang (huangky@ihb.ac.cn) or M. Hippler (mhippler@uni-muenster.de).

## Acknowledgements

The work in the laboratory of KH was supported by the National Nature Science Foundation of China (Grant 31671399 to Huang K). MH acknowledges support from the German Science Foundation (DFG, HI739/12-1). LNL acknowledges funding support from the Royal Society (UF120411, URF\R\180030, RGF\EA\181061 and RGF\EA\180233) and the Biotechnology and Biological Sciences Research Council (BB/R003890/1, BB/M012441/1). China Postdoctoral Science Foundation Funded Project (Project No.: 2019M662335). AO and JB thank S Schulze for providing SugarPy. AG, MK and OB thank M Lorenz and the Göttingen Algae Culture Collection (SAG) for providing the WT-SAG strain and R Catalan for technical assistance. Thanks, Z Liang, J Huang and Professor K Jiang at Wuhan University for sharing the TRIF microscope. Thanks D Tan and Bo Zhu and Professor L Xue at Wuhan University for sharing the AFM microscope and the technique assistance.

## Additional information

### Funding

| Funder | Grant reference number | Author |
| --- | --- | --- |
| Deutsche Forschungsgemeinschaft | HI737/12-1 | Michael Hippler |
| National Natural Science Foundation of China | 31671399 | Kaiyao Huang |
| Royal Society | UF120411 | Lu-Ning Liu |
| Biotechnology and Biological Sciences Research Council | BB/R003890/1 | Lu-Ning Liu |
| Biotechnology and Biological Sciences Research Council | BB/M012441/1 | Lu-Ning Liu |
| Royal Society | URF\R\180030 | Lu-Ning Liu |
| Royal Society | RGF\EA\181061 | Lu-Ning Liu |

| Royal Society | RGF\EA\180233 | Lu-Ning Liu |
| China Postdoctoral Science Foundation | 2019M662335 | Longsheng Zhao |

The funders had no role in study design, data collection and interpretation, or the decision to submit the work for publication.

### Author contributions

Nannan Xu, Lara Hoepfner, Investigation, Writing - review and editing; Anne Oltmanns, Formal analysis, Investigation, Writing - original draft; Longsheng Zhao, Antoine Girot, Marzieh Karimi, Simon Kelterborn, Investigation; Martin Scholz, Formal analysis; Julia Beißel, Formal analysis, Investigation; Peter Hegemann, Supervision, Writing - review and editing; Oliver Bäumchen, Funding acquisition, Validation, Writing - review and editing; Lu-Ning Liu, Supervision, Funding acquisition, Writing - review and editing; Kaiyao Huang, Michael Hippler, Conceptualization, Supervision, Funding acquisition, Project administration, Writing - review and editing

### Author ORCIDs

Nannan Xu (iD) https://orcid.org/0000-0003-2847-7225
Anne Oltmanns (iD) https://orcid.org/0000-0002-0153-048X
Lara Hoepfner (iD) https://orcid.org/0000-0002-8222-4563
Peter Hegemann (iD) http://orcid.org/0000-0003-3589-6452
Oliver Bäumchen (iD) http://orcid.org/0000-0002-4879-0369
Lu-Ning Liu (iD) http://orcid.org/0000-0002-8884-4819
Kaiyao Huang (iD) https://orcid.org/0000-0001-8669-1065
Michael Hippler (iD) https://orcid.org/0000-0001-9670-6101

### Decision letter and Author response

Decision letter https://doi.org/10.7554/eLife.58805.sa1
Author response https://doi.org/10.7554/eLife.58805.sa2

## Additional files

### Supplementary files

• Transparent reporting form

### Data availability

The mass spectrometry proteomics data (Figure 1) have been deposited to the ProteomeXchange Consortium (http://proteomecentral.proteomexchange.org) via the PRIDE partner repository with the dataset identifier PXD018353.

The following dataset was generated:

| Author(s) | Year | Dataset title | Dataset URL | Database and Identifier |
| --- | --- | --- | --- | --- |
| Oltmanns A, Beißel J, Scholz M, Hippler M | 2020 | Figure 1 data | https://www.ebi.ac.uk/pride/archive/projects/PXD018353 | PRIDE, PXD018353 |

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
