## [Decision Letter]

**Acceptance summary:**

This article examines the basis for the adhesion of the bi-flagellate cell *Chlamydomonas* to substrates, which is important for the ability of these cells to live in diverse habitats. Substrate adhesion in *Chlamydomonas* is due to flagellar membrane protein FMG-1B. This paper shows that mutations that disrupt *N*-glycosylation of FMG-1B do not change its abundance and flagellar localization, but reduces the strength of cell adhesion to solid substrates.

**Decision letter after peer review:**

Thank you for submitting your article "Altered *N*-glycan composition impacts flagella mediated adhesion in *Chlamydomonas reinhardtii* " for consideration by *eLife*. Your article has been reviewed by three peer reviewers, and the evaluation has been overseen by a Reviewing Editor and Piali Sengupta as the Senior Editor. The following individuals involved in the review of your submission have agreed to reveal their identity: Robert Bloodgood (Reviewer #1); Felix Rico (Reviewer #2).

The reviewers have discussed the reviews with one another and the Reviewing Editor has drafted this decision to help you prepare a revised submission.

Summary:

This article examines the basis for the adhesion of the bi-flagellate cell *Chlamydomonas* to substrates. Substrate adhesion in *Chlamydomonas* is due to the flagellar surface. FMG-1B constitutes most of the protein of the *Chlamydomonas* flagellar membrane and is the major surface-exposed flagellar membrane protein and the only flagellar membrane protein that contacts a planar substrate during whole-cell gliding motility or a polystyrene microsphere during the other manifestation of surface motility associated with the flagellum. The authors are asking the rather straight forward question of whether alterations in the *N*-glycosylation of the flagellar membrane glycoproteins affects the strength of cell adhesion of *Chlamydomonas* with a substrate. The study uses two previously characterized insertional mutants, IM*_Man1A_* ( in mannosidase) and IM*_XylT1A_* (in xylosyltransferase 1A), a corresponding double mutant and a novel CRISPR mutant (CRISPR*_XylT1A_1_* also in xylosyltransferase 1A) to analyze how changes in protein *N*-glycosylation affect flagellar adhesion to surfaces using *Chlamydomonas*.

The results show that altered *N*-glycosylation of FMG-1B does not change its abundance and flagellar localization, or the IFT and gliding velocity of the cells. However, it affects the binding of polystyrene microbeads to the flagella, as well as its transport. Moreover, AFM measurements show a decrease of both the forces and average energy in the mutants in comparison to the WT strain. Besides, micropipette force measurements performed on strains whose *N*-glycan composition was altered using the CRISPR approach showed also a decrease of the adhesion force on mutant cells under blue light (known to enhance adhesion of the WT reference). Finally, measurements done with and without ciliobrevin D show that ciliobrevin D is not increasing the adhesion forces of mutant cells contrary to what is observed for WT cells.

While the reviewers agreed that perturbations in the *N*-glycosylation pattern affect the level of cell adhesion and can be an interesting result worthy of being reported in *eLife*, they also noted major weaknesses and raised serious concerns about the validity of the conclusions made in the manuscript and the advance the manuscript makes in the field. Specifically, the key AFM and micropipette experiments, while very elegant and of great potential, as shown in a previous publication, were performed in the presence of a drug, which appears to kill and damage many cells. It remains unclear whether the changes in glycosylation directly impact adhesion or change protein folding or cilia protein composition. Also, the mutations do not eliminate *N*-glycosylation, but alter the composition and size, raising the question of how these changes would interfere with adhesion, considering that the double mutant shows subtle changes in anti-HRP staining. Thus, the panel recommended that the work in its present form is too preliminary to be considered for publication in *eLife*, and the manuscript could benefit greatly by a major revision. The major points listed below provide clear guidance as to how to improve the manuscript.

Essential revisions:

1) It appears that the difference in "Adhesion Force" between control and the *_xylT1A_*mutant are different in Figure 4A and 3C – this might be due to the differences in the method. However, since different strains were used, it is also possible that there might be a large strain-specific difference in the adhesion phenotype (two distinct wild-type strains were used) that are not necessarily related to the mutations of interest. Since no rescue experiments are presented, the correlation between the genotype and the quite variable phenotypes is somewhat uncertain.

2) The pattern obtained by staining of flagella extracts of the double mutant with anti-HRP is quite similar to that of wild-type flagella. How does that fit with the observation that adhesion forces are most affected in the double mutant? Concerning protein glycosylation, the double mutant is better off than the two single mutants (e.g. binding of anti-HRP is reduced in the single mutant but returned to normal in the double) but no explanation is offered why that is the case. Schulze et al. state that loss of mannosidase suppresses the XlyT1A mutation but – since it is called a knock-down mutant -it appears that the Man1A mutant is not null?

3) The authors observe that Ciliobrevin D (an inhibitor of IFT dynein) treatment increases the adhesion force. However, as stated in the Materials and methods section, many cells died during the treatment of cells with ciliobrevin D. Thus, it is unclear whether the effect observed here is really to the inhibition of dynein or due to some toxicity of the drug.

"The impact of ciliobrevin D on the adhesion force suggested that IFT might contribute to surface adhesion forces." But differences in IFT were not observed. So, how do the authors explain the effect of ciliobrevin? Does active gliding make adhesion less strong?

Since ciliobrevin D is believed to inhibit IFT dynein, the panel recommends using strains that abolish IFT by a temperature shift such as fla10ts to further explore whether active IFT lowers cell adhesion.

4) The authors should discuss the nature and the relevance of these binding measurements. What is the natural surface of binding for this organism in real life? While polystyrene (beads), glass (AFM) and silicon (micropipette) present all similar surface charges, what kind of interaction is reported in this work? It would be useful to provide a plausible description of the interaction between glycans and the surfaces.

5) The manuscript is complicated, difficult to read and the authors could do better with explaining the logic behind some of their experiments. To guide the reader through the numerous variations, some details should be more explicitly set out. In particular, the last part of the discussion, regarding the impact of ciliobrevin D and its link with gliding, can probably be explained more clearly and further discussed. Some details are currently lacking to give a clear overview.

[Editors' note: further revisions were suggested prior to acceptance, as described below.]

Thank you for submitting your article "Altered *N*-glycan composition impacts flagella mediated adhesion in *Chlamydomonas reinhardtii* " for consideration by *eLife*. Your article has been reviewed by three peer reviewers, and the evaluation has been overseen by a Reviewing Editor and Piali Sengupta as the Senior Editor. The following individuals involved in review of your submission have agreed to reveal their identity: Robert Bloodgood (Reviewer #1); Felix Rico (Reviewer #2).

The reviewers have discussed the reviews with one another and the Reviewing Editor has drafted this decision to help you prepare a revised submission.

Summary:

The authors have made a commendable effort to address the many comments and requests of the reviewers, and have performed new experiments to respond to the reviews. This has resulted in a greatly improved manuscript which will be much easier for readers to understand. This paper makes a worthwhile contribution to the field of cell adhesion (in particular flagellar adhesion) and is likely to be of interest to many readers of *eLife* provided that they can revise their manuscript in response to the remaining concerns below.

Revisions:

1) "reduction in *N*-glycan complexity impedes the adhesion force required for binding the flagella to surfaces as demonstrated by force spectroscopy and impairs polystyrene bead binding and transport"

Delete "as demonstrated by force spectroscopy" as the beginning of this sentence explains the method used. It is important to clarify that "…reduction in *N*-glycosylation reduces the flagellar adhesion force. This results in a reduction of bead motility but not gliding of cells on solid surfaces."

2) "Since (the) retrograde motor dynein-1b pauses relative to the adhesion…"

According to Shih et al., it is the retrograde IFT trains that pause at the adhesion sites and dynein-1b motors carrying these paused trains pull the axoneme in the opposite direction.

3) "Considering that *N*-glycoproteins per se are important for adhesion but are constantly lost from the flagellar membrane, gliding is supposed to require an enormous amount of energy, which suggests that flagella mediated adhesion has a somewhat high importance. Furthermore, it opens the question whether the maturation of *N*-glycans (as additional energy expense) in Golgi is important for flagella mediated adhesion, i.e. whether *N*-glycosylation is crucial for adhesion beyond proper glycoprotein folding."

The reasoning with ATP costs are rather abstract and not relevant. This section should be removed from the manuscript. Instead, the authors could point out in the Introduction that *Chlamydomonas* species often grow on solid surfaces (and then might move by gliding) rather than living in aquatic habitats. Because many *Chlamydomonas* species can live on a wide variety of surfaces (soil, sand, wet leaves, moss, and bark), the flagellar surface adhesion system has not developed high specificity but rather has great flexibility.

4) "two insertional mutants such as IM*_Man1A_*, IM*_XylT1A_* and their double mutant IM*_Man1A_*xIM*_XylT1A_* were studied."

Define IM as in insertional mutants here.

5) "that altered *N*-glycan maturation did not affect the flagellar localization of FMG-1B."

The authors should cite Bloodgood, Salomonsky and Reinhart, 1987, Figure 5 of this paper has the same result shown in Figure 1 of the manuscript. The paper also reported that the glycosylation defective FMG-1B mutant exhibited increased binding of Concanavalin A.

6) "As it cannot be excluded that flagella shortening, as induced upon temperature shift (Figure 5—figure supplement 1D), impacted flagellar adhesion forces, the triple mutant treated with 20 mM NaPPi was assessed by AFM as control."

It is unclear why NaPPi treatment was used as a control. Because flagellar shortening can't be excluded? Was a flagellar length-dependency of the adhesion force observed?

7) "It could be speculated, that the action of dynein-1b reduces flagellar adhesion forces due to its opposite direction of action with respect to the flagellar adhesion force."

It is unclear what the authors mean by "opposite direction of action". Does this mean that IFT dynein pulls the membrane away from the substrate? Or, do forces generated by dynein motors on paused IFT trains dissociate the FMG1B clusters from the surface, effectively reducing the flagellar adhesion forces?

8) "Of note, also FMG-1A is localized in flagella (but) and its abundance was unaltered between WT and mutants in vegetative cells (Figure 1—figure supplement 4)."

It is not clear what is concluded here.

9) "Our data further suggested that flagellar assembly…".

The authors should more clearly state that *N*-glycosylation is not needed for flagellar elongation, IFT, and FMG-1B, but it is needed for adhesion of flagellar surface to the solid substrates and that the diminished adhesion force in *N*-glycan mutants is still sufficient for gliding. We recommend deleting "This suggests that the evolution of adhesion might not have been governed only by the capability of gliding but by the necessity of adhesion itself."

10) "Given the response of flagellar adhesion to blue light, it could potentially link adhesion to photo-protection which is also blue-light-mediated, as adhesion might result in photoprotection via cell shading"

It is unclear what the authors mean by shading here. Just because a cell (the width of flagella is smaller than the wavelength of blue light) adheres does not make any shade. This either needs to be elaborated or removed.

11) Figure 1—figure supplement 3: Showing just one half of the lectin blot is sufficient.

---

## [Author Response]

While the reviewers agreed that perturbations in the *N*-glycosylation pattern affect the level of cell adhesion and can be an interesting result worthy of being reported in eLife, they also noted major weaknesses and raised serious concerns about the validity of the conclusions made in the manuscript and the advance the manuscript makes in the field. Specifically, the key AFM and micropipette experiments, while very elegant and of great potential, as shown in a previous publication, were performed in the presence of a drug, which appears to kill and damage many cells. It remains unclear whether the changes in glycosylation directly impact adhesion or change protein folding or cilia protein composition. Also, the mutations do not eliminate N-glycosylation, but alter the composition and size, raising the question of how these changes would interfere with adhesion, considering that the double mutant shows subtle changes in anti-HRP staining. Thus, the panel recommended that the work in its present form is too preliminary to be considered for publication in eLife, and the manuscript could benefit greatly by a major revision. The major points listed below provide clear guidance as to how to improve the manuscript.

The HRP blot has been repeated, revealing clearer differences between WT and *N*-glycosylation mutants. Additionally, Concanavalin A affino blotting has been performed illustrating increased Concanavalin A affinity for all *N*-glycosylation mutants, supporting altered *N*-glycan maturation in all mutants as compared to WT as already observed by mass spectrometry. Immuno-fluorescence images have further been repeated in a simplified approach to allow better comprehension. To address the important issue of comparability of adhesion forces measured via the two different approaches (AFM and micropipette force measurement) adhesion forces have now also been assessed in presence of ciliobrevin D in the micropipette approach. Finally, we present experimental evidence that ciliobrevin D acts on dynein and that the forces measured are not inflicted by its putative toxicity. To do so, a temperature sensitive mutation of cytoplasmic dynein was introduced in the double mutant IM*_Man1A_*XIM*_XylT1A_* and consequently, the adhesion force of this mutant was measured at permissive and restrictive temperatures via AFM as suggested by the reviewers.

Essential revisions:1) It appears that the difference in "Adhesion Force" between control and the xylT1A mutant are different in Figure 4A and 3C – this might be due to the differences in the method. However, since different strains were used, it is also possible that there might be a large strain-specific difference in the adhesion phenotype (two distinct wild-type strains were used) that are not necessarily related to the mutations of interest. Since no rescue experiments are presented, the correlation between the genotype and the quite variable phenotypes is somewhat uncertain.

Indeed, the measured adhesion force possesses a strain specific component. This is an interesting finding, yet beyond the scope of our manuscript. To further support the finding that ciliobrevin D impacts the adhesion forces measured and to allow a better comparison of both methods used for adhesion force determination, micropipette force measurements have been additionally performed in presence of ciliobrevin D as employed for AFM measurements (new Figure 4). These measurements confirmed that the effect of ciliobrevin D can be observed in both measurements. Thus, independent of genetic wildtype background and type of adhesion force measurement, we observed that changes in *N*-glycosylation composition impact adhesion to surface. In our eyes, this underpins the robustness of our results (despite varying strains and methods) and even strengthens the universality of our finding.

2) The pattern obtained by staining of flagella extracts of the double mutant with anti-HRP is quite similar to that of wild-type flagella. How does that fit with the observation that adhesion forces are most affected in the double mutant? Concerning protein glycosylation, the double mutant is better off than the two single mutants (e.g. binding of anti-HRP is reduced in the single mutant but returned to normal in the double) but no explanation is offered why that is the case. Schulze et al. state that loss of mannosidase suppresses the XlyT1A mutation but – since it is called a knock-down mutant -it appears that the Man1A mutant is not null?

For now, we do not have molecular insights of how altered *N*-glycosylation impacts surface adhesion, i.e. which *N*-glycosylated proteins are involved and which *N*-glycan components are crucial for “WT-like” action. This question will be very interesting to address in future research. To avoid extensive, not data-supported speculations, we omitted the discussion on why the double mutant is affected most. We see that we did not state this clearly enough in the previous version of the manuscript and now included a respective paragraph in the revised version. Further, to show clearly that all mutants analysed in this manuscript show specific alterations in their *N*-glycan patterns, the HRP-blot has been repeated and a Concanavalin A affino blotting has been performed in addition. The HRP-blot now clearly revealed increased binding of HRP in both IM*_Man1A_* and double mutant as well as reduced binding in IM*_XylT1A_* compared to WT-Ins (Figure 1—figure supplement 2). Additionally, the Con-A affino blot shows increased affinity of Con-A in all mutants analyzed (Figure 1—figure supplement 3). These results now clearly provide evidence that also the double mutant shows altered *N*-glycan maturation and is not “better off” regarding *N*-glycosylation.

3) The authors observe that Ciliobrevin D (an inhibitor of IFT dynein) treatment increases the adhesion force. However, as stated in the Materials and methods section, many cells died during the treatment of cells with ciliobrevin D. Thus, it is unclear whether the effect observed here is really to the inhibition of dynein or due to some toxicity of the drug.

We apologize that our wording in the previous version of the manuscript was not clear causing a misunderstanding. We do not observe any toxicity effect of the drug (ciliobrevin D) during preparation. Our intention of this technical remark was to highlight the effect of centrifugation during preparation to separate alive from dead or deflagellated cells. To pipette only the swimming cells, only the upper part of the treated cell suspension (about 15 mL) was used for these experiments. No dead cells are observed in the liquid chamber during experiments. The observed effect is truly related to the inhibition of dynein. We have thus revised the manuscript accordingly.

"The impact of ciliobrevin D on the adhesion force suggested that IFT might contribute to surface adhesion forces." But differences in IFT were not observed. So, how do the authors explain the effect of ciliobrevin? Does active gliding make adhesion less strong?Since ciliobrevin D is believed to inhibit IFT dynein, the panel recommends using strains that abolish IFT by a temperature shift such as fla10ts to further explore whether active IFT lowers cell adhesion.

Thanks for the reviewer’s insightful comments. To rule out the toxicity of ciliobrevin D, we generated the triple mutant dynein-1b^ts^ XIM*_Man1A_*XIM_*XylT1A*-4-13#_ (double mutant crossed with CC-4423, a dynein-1b^ts^ mutant, in which the retrograde IFT stop and flagella disassemble at restrictive temperature) (Engel, Ishikawa et al., 2012). We added these data in Figure 5—figure supplement 1. In order to exclude the effect of flagellar length to adhesion force, we compared the force measured for dynein-1b^ts^XIM*_Man1A_*XIM_*XylT1A*-4-13#_ at restrictive temperature with the double mutant IM*_Man1A_*XIM*_XylT1A_* treated with NaPPi. The treatment of NaPPi has no effect on the IFT (Dentler, 2005). In line with elevated adhesion forces in the presence of ciliobrevin D, the new triple mutant showed elevated adhesion forces at restrictive temperature (as compared to the NaPPi control sample). These results demonstrated on one hand, that the effect assessed in the presence of ciliobrevin D is not attributed to toxicity and on the other, it indicates that active dynein-1b decreases flagellar adhesion forces (see Figure 5—figure supplement 1). Of course, this is an interesting finding opening a variety of questions to be answered in the future, however, showing how exactly IFT or dynein 1-b are involved in adhesion is not the intention of this manuscript which instead focuses on the question if glycosylation impacts adhesion.

4) The authors should discuss the nature and the relevance of these binding measurements. What is the natural surface of binding for this organism in real life? While polystyrene (beads), glass (AFM) and silicon (micropipette) present all similar surface charges, what kind of interaction is reported in this work? It would be useful to provide a plausible description of the interaction between glycans and the surfaces.

We agree that it will be very interesting to analyze the precise interaction of *N*-glycan and surface. As described in Kreis et al., 2019, flagella-mediated adhesion is largely independent of substrate properties. It is therefore at the moment not possible to define the interaction of glycans and surface observed in this study. Statements regarding this question would be purely speculative.

5) The manuscript is complicated, difficult to read and the authors could do better with explaining the logic behind some of their experiments. To guide the reader through the numerous variations, some details should be more explicitly set out. In particular, the last part of the discussion, regarding the impact of ciliobrevin D and its link with gliding, can probably be explained more clearly and further discussed. Some details are currently lacking to give a clear overview.

Thanks. The manuscript was rewritten to be more comprehensive.

[Editors' note: further revisions were suggested prior to acceptance, as described below.]

Revisions:1) "reduction in *N*-glycan complexity impedes the adhesion force required for binding the flagella to surfaces as demonstrated by force spectroscopy and impairs polystyrene bead binding and transport"Delete "as demonstrated by force spectroscopy" as the beginning of this sentence explains the method used. It is important to clarify that "…reduction in *N*-glycosylation reduces the flagellar adhesion force. This results in a reduction of bead motility but not gliding of cells on solid surfaces."

Thank you for the note. We have deleted the suggested part of the sentence to avoid repetition and revised the paragraph as follows:

“Taking advantage of atomic force microscopy and micropipette force measurements, our data revealed that reduction in *N*-glycan complexity impedes the adhesion force required for binding the flagella to surfaces. This results in impaired polystyrene bead binding and transport but not gliding of cells on solid surfaces.”

2) "Since (the) retrograde motor dynein-1b pauses relative to the adhesion…"According to Shih et al., it is the retrograde IFT trains that pause at the adhesion sites and dynein-1b motors carrying these paused trains pull the axoneme in the opposite direction.

We have corrected the paragraph as follows:

“Since retrograde IFT trains pause relative to the adhesion site while FMG-1B tethers to the solid surface through its large extracellular carbohydrate domain (Bloodgood, 2009), the force generated by retrograde motor protein dynein-1b will push the microtubule into the opposite direction, dragging the cell body and the second flagellum behind; the gliding process is initiated (Shih et al., 2013).”

3) "Considering that *N*-glycoproteins per se are important for adhesion but are constantly lost from the flagellar membrane, gliding is supposed to require an enormous amount of energy, which suggests that flagella mediated adhesion has a somewhat high importance. Furthermore, it opens the question whether the maturation of *N*-glycans (as additional energy expense) in Golgi is important for flagella mediated adhesion, i.e. whether *N*-glycosylation is crucial for adhesion beyond proper glycoprotein folding."The reasoning with ATP costs are rather abstract and not relevant. This section should be removed from the manuscript. Instead, the authors could point out in the Introduction that Chlamydomonas species often grow on solid surfaces (and then might move by gliding) rather than living in aquatic habitats. Because many Chlamydomonas species can live on a wide variety of surfaces (soil, sand, wet leaves, moss, and bark), the flagellar surface adhesion system has not developed high specificity but rather has great flexibility.

We have revised the paragraph and added the suggestions to the Introduction:

“Due to the high *N*-glycosylation level of the extraflagellar domain of FMG1-B, interacting with the solid surface, it was suggested that *N*-glycosylation could be crucial for adhesion, beyond proper glycoprotein folding.”

“These findings imply that the adhesion system of *C. reinhardtii* has developed toward great flexibility instead of high specificity, in line with the high diversity of solid surfaces dwelled by the microalga in nature ranging from soil and sand to wet leaves, moss and bark (Harris, 2009).”

4) "two insertional mutants such as IMMan1A, IMXylT1A and their double mutant IMMan1AxIMXylT1A were studied."Define IM as in insertional mutants here.

The abbreviation is now defined in the text.

5) "that altered N-glycan maturation did not affect the flagellar localization of FMG-1B."The authors should cite Bloodgood, Salomonsky and Reinhart, 1987, Figure 5 of this paper has the same result shown in Figure 1 of the manuscript. The paper also reported that the glycosylation defective FMG-1B mutant exhibited increased binding of Concanavalin A.

Thank you for the note. We agree and have added respective reference and discussion:

“This is in line with early findings by Bloodgood et al., reporting proper FMG-1B targeting to the flagella in a *N*-glycosylation mutant termed L23 showing increased ConA-affinity (Bloodgood et al., 1987).”

6) "As it cannot be excluded that flagella shortening, as induced upon temperature shift (Figure 5—figure supplement 1D), impacted flagellar adhesion forces, the triple mutant treated with 20 mM NaPPi was assessed by AFM as control."It is unclear why NaPPi treatment was used as a control. Because flagellar shortening can't be excluded? Was a flagellar length-dependency of the adhesion force observed?

As we did not focus on the question whether differences in flagella length influence flagellar adhesion forces, no experiments of such kind were performed. Since we, however, cannot exclude that there might be a dependency, we decided to analyze cells via AFM with a similar flagella length of about 6.5 μm. Therefore, we used NaPPi to shorten the flagella to 6.5 μm in the tripe mutant at non-restrictive temperature (after 20 min of treatment), a length that was observed at restrictive temperatures in the triple mutant. This has been now clarified in manuscript and the figure legend.

7) "It could be speculated, that the action of dynein-1b reduces flagellar adhesion forces due to its opposite direction of action with respect to the flagellar adhesion force."It is unclear what the authors mean by "opposite direction of action". Does this mean that IFT dynein pulls the membrane away from the substrate? Or, do forces generated by dynein motors on paused IFT trains dissociate the FMG1B clusters from the surface, effectively reducing the flagellar adhesion forces?

We clearly do not know why and how exactly the action of dynein-1b results in increased adhesion force. We suggest that this is due to a destabilization of the adhering protein cluster via dynein-1b motility resulting in decreased adhesion force while dynein-1b is active. We have revised the respective paragraph and hope to have presented our hypothesis more clearly.

8) "Of note, also FMG-1A is localized in flagella (but) and its abundance was unaltered between WT and mutants in vegetative cells (Figure 1—figure supplement 4)."It is not clear what is concluded here.

Indeed, we did not make our intentions clear enough in the cited lines. So far very little is known on the role of FMG-1A, often only referred to as minor (unimportant) form of FMG-1. It has been proposed to be mainly expressed in reproductive cells. In our mass spectrometry data, however, FMG1-A is highly abundant also in flagella of vegetative cells indicating that it might be more important than suggested until now.

We have revised the paragraph as follows:

“Of note, also FMG-1A is localized in flagella and its abundance was unaltered between WT and mutants in vegetative cells (Figure 1—figure supplement 4). This contrasts the current assumption that FMG-1A is solely expressed in reproductive cells and opens the question whether it might have a similar role as FMG-1B, given the high similarity of the two proteins (Bloodgood, 2009).”

9) "Our data further suggested that flagellar assembly…".The authors should more clearly state that N-glycosylation is not needed for flagellar elongation, IFT, and FMG-1B, but it is needed for adhesion of flagellar surface to the solid substrates and that the diminished adhesion force in N-glycan mutants is still sufficient for gliding. We recommend deleting "This suggests that the evolution of adhesion might not have been governed only by the capability of gliding but by the necessity of adhesion itself."

We have performed the recommended deletion and revised the paragraph as follows:

“Our data further suggested that flagellar assembly, IFT and FMG-1B transport into flagella were not affected by altered *N*-glycosylation implicating no role of *N*-glycosylation in these processes. Instead, proper *N*-glycosylation of flagellar proteins is crucial for adhering *C. reinhardtii* cells onto surfaces. Our observations further suggest that the remaining adhesion force, although diminished in *N*-glycan mutants, is still sufficient for gliding.”

10) "Given the response of flagellar adhesion to blue light, it could potentially link adhesion to photo-protection which is also blue-light-mediated, as adhesion might result in photoprotection via cell shading"It is unclear what the authors mean by shading here. Just because a cell (the width of flagella is smaller than the wavelength of blue light) adheres does not make any shade. This either needs to be elaborated or removed.

We agree that our thoughts were not stated clearly enough. Our hypothesis is that in response to blue-light *C. reinhardtii* might form multi-layered biofilms by adhering to a surface. This biofilm might thereby allow mutual cell shading leading to photoprotection. This has now been clarified in the manuscript as follows:

“Given the response of flagellar adhesion to blue light, it could potentially link adhesion to photo-protection which is also blue-light mediated, as adhesion might result in photoprotection via biofilm formation, which in turn would enable mutual cell shading.”

11) Figure 1—figure supplement 3: Showing just one half of the lectin blot is sufficient.

The figure has been changed as recommended.